# Towards Interpretable, Sequential Multiple Instance Learning: An Application to Clinical Imaging

## Abstract

This work introduces the Sequential Multiple Instance Learning (SMIL) framework, addressing the challenge of interpreting sequential, variable-length sequences of medical images with a single diagnostic label. Diverging from traditional MIL approaches that treat image sequences as unordered sets, SMIL systematically integrates the sequential nature of clinical imaging. We develop a bidirectional Transformer architecture, BiSMIL, that optimizes for both early and final prediction accuracies through a novel training procedure to balance diagnostic accuracy with operational efficiency. We evaluate BiSMIL on three medical image datasets to demonstrate that it simultaneously achieves state-of-the-art final accuracy and superior performance in early prediction accuracy, requiring 30-50% fewer images for a similar level of performance compared to existing models. Additionally, we introduce SMILU, an interpretable uncertainty metric that outperforms traditional metrics in identifying challenging instances.

## 1 Introduction

Medical imaging is a fundamental component of modern medical diagnosis. With the surge in availability of imaging, there has been widespread interest in leveraging computer vision techniques to aid interpretation of medical images.

A common challenge in medical imaging is that each image study often contains multiple number of image instances, with only one associated diagnostic label. The sequence length can also vary significantly across patients, making conventional deep learning models, largely tailored for fixed input sizes, inappropriate.

To solve this problem, there has been a growing literature to develop Multiple Instance Learning (MIL) methods that can tackle this setting. There has been significant work in developing MIL methods for whole slide images (Courtiol et al., 2018; Campanella et al., 2019; Shao et al., 2021b; Li et al., 2021; Lu et al., 2021; Zhang et al., 2022; Liu et al., 2023), where the set of images are often treated as an order-independent set denoted as a "bag". In many clinical imaging settings, however, clinicians are creating the images sequentially to discover features of interest. They often have control over how many sequential images should be created (e.g. CT scan levels) or when to stop the sequential imaging process (e.g. ultrasound). This sequential nature is currently largely ignored in applications of MIL to clinical imaging (Ostrowski et al., 2023; Fuhrman et al., 2023).

In this work, we present a Sequential MIL (SMIL) framework that aims to systematically incorporate the sequential nature of clinical imaging into MIL. In particular, the sequential nature of clinical imaging generates a unique tradeoff between accuracy and efficiency: as the clinician creates more images, the resulting diagnostic accuracy is likely to increase, but it comes at the expense of efficiency and patient radiation exposure. Therefore, in the SMIL framework, it is critical to develop methods that can achieve accurate predictions in an early subsequence. However, existing medical imaging datasets most often do not have labels for subsequences, and the bag-level label might not be correct for the subsequence, making training difficult.

To tackle the SMIL framework, we formulate a new bidirectional Transformer architecture, BiSMIL, that exploits the sequential nature of clinical images. We further develop a novel training procedure

for the BiSMIL model that encourages the model to give an accurate early prediction while ensuring it has a high final accuracy.

We evaluate the BiSMIL model on three independent medical image datasets, including a new dataset on ultrasounds for pediatric urology, where the sonographer has full control over the number of images he/she wishes to create to classify urinary tract dilation. We demonstrate that the BiSMIL model is able to consistently outperform existing approaches in both final prediction accuracy and early prediction accuracy. Importantly, the BiSMIL model can achieve high early prediction accuracy with 40%-60% fewer instances compared to existing models.

To further the applicability of the model, we also develop an interpretable, sequence-aware uncertainty metric SMILU that allows clinicians to understand the certainty of the SMIL prediction. SMILU depends not only on its final prediction, but also the incremental predictions over the sequence of images. Our experiments demonstrate that the SMILU metric is able to better capture difficult-to-classify, uncertain instances better than common metrics that are based solely on the final output.

In summary, our contributions are three-fold:

- We introduce the SMIL framework that systematically incorporates the sequential structure of clinical imaging into MIL. The SMIL framework exhibits unique challenges for MIL methods to provide early, accurate predictions without access to subsequence labels.
- We propose a bidirectional transformer architecture, BiSMIL, to tackle the SMIL framework, and formulate a novel training procedure to reliably encourage accurate early predictions while still ensuring high accuracy for final predictions.
- We provide an interpretable, sequence-aware uncertainty metric SMILU that allows clinicians to understand the certainty of the SMIL prediction. We show that the uncertainty metric outperforms common metrics in recognizing uncertain, difficult-to-classify instances in the SMIL framework.

## 2 RELATED WORK

**Multiple Instance Learning (MIL).** Multiple Instance Learning (MIL) is a weakly supervised learning framework, wherein instances are grouped into bags with labels designated at the bag level (Dieterich et al., 1997; Ramon and De Raedt, 2000; Andrews et al., 2002; Settles et al., 2007; Li and Vasconcelos, 2015; Ilse et al., 2018). There has been a particularly high level of interest in utilizing the MIL framework for histopathology slides which possesses high resolutions up to $10^5 \times 10^5$. To address the issue of training neural networks on such images, each slide is commonly divided into hundreds or thousands of tiles, and the MIL framework has been widely developed and utilized for this application (Courtiol et al., 2018; Campanella et al., 2019; Shao et al., 2021a;b; Li et al., 2021; Lu et al., 2021; Zhang et al., 2022; Liu et al., 2023). However, this means that traditionally MIL assumes an absence of sequential interactions between instances. The few works that do capture relationships between instances within a bag (Zhou et al., 2009; Tu et al., 2019; Wu et al., 2023) do not systematically consider the sequential nature of clinical imaging.

**Interpretability for MIL.** There has been significant work in enhancing the interpretability of MIL methods. Most work has focused on identifying particular instances in a bag that contribute significantly to the final prediction (Pirovano et al., 2020; Wang et al., 2019a; Javed et al., 2022; Ilse et al., 2018; Molnar, 2020; Early et al., 2022). Our focus diverges from these works as we aim to provide a bag-level metric that signals the certainty of the MIL model in predicting a particular bag.

## 3 METHODS

In this section, we provide an overview of the general SMIL framework, propose the specific BiSMIL model, and introduce the novel training procedure that we will utilize to train the BiSMIL model.

### 3.1 SEQUENTIAL MULTIPLE INSTANCE LEARNING

In the classical Multiple Instance Learning (MIL) setting, the dataset is represented by a collection of bags, $\{\mathbf{X}_1, \mathbf{X}_2, \ldots, \mathbf{X}_n\}$, where each bag $\mathbf{X}_i \in \mathcal{X}$ contains $m_i$ instances $\{\mathbf{x}_{i1}, \mathbf{x}_{i2}, \ldots, \mathbf{x}_{im_i}\}$. In

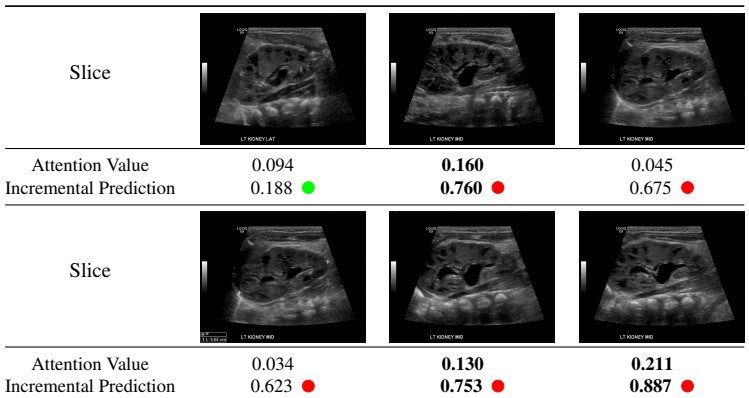

Figure 1: An illustration of incremental predictions for the first 6 instances in a particular sequence of the UTD dataset along with the instance-level attention values. Green indicates a negative prediction and red represents a positive prediction. Instances with the highest attention values are bolded.

the common scenario where each instance corresponds to an image, we have $\mathbf{x}_{ij} \in \mathbb{R}^{l \times w}$. Each bag $\mathbf{X}_i$ is associated with a binary label $y_i \in \{0, 1\}$, where $y_i = 0$ if all instances are negative and $y_i = 1$ if any instance is positive. The goal of classic MIL is to learn a machine learning model parametrized by $\boldsymbol{\theta}$, $f_{\boldsymbol{\theta}} : \mathcal{X} \to \{0, 1\}$ that can accurately learn the bag-level labels across bags that have varying number of instances $m_i$.

In the Sequential MIL (SMIL) Framework, instances within each bag $i$ are generated sequentially, implying an associated time $t_{ij}$ for each instance $\mathbf{x}_{ij}$, with $t_{ij} < t_{ik}$ for all $j < k$ and $i \in [n]$. Therefore, we denote $\mathbf{X}_i$ as a *sequence* rather than a bag to emphasize this temporal dependance. The aim is to provide a model $f$ that, upon the generation of the $j$-th image, offers an *incremental prediction* $p_{ij} = f(\mathbf{X}_i^j) \in [0, 1]$, reflecting the likelihood that the current subsequence $\mathbf{X}_i^j = \{\mathbf{x}_{i1}, \ldots, \mathbf{x}_{ij}\}$ warrants a positive diagnosis. An accurate incremental prediction $p_{ij}$ can facilitate clinicians to make informed decisions on whether to change or terminate the imaging sequence. This can improve clinical efficiency and reduce radiation exposure.

Thus, in the SMIL framework, accurate, early incremental predictions are crucial for a successful model. To further illustrate this concept, in Figure 2, we showcase the incremental prediction results for a positive sample ($y_i = 1$) from one of the medical imaging datasets. Notably, we observe that the prediction values indicate the 2nd, 5th, and 6th image appears to have significantly contributed to a positive prediction. We note that these corresponded with a high attention value to these images, and importantly corresponded with evaluation from a clinician, who stated that only these three images showed any signs of abnormality. Given the three strong incremental predictions, the clinician can arguably stop the imaging sequence after the 6th instance to improve operational efficiency.

However, Figure 2 also highlights a fundamental challenge within the SMIL framework: Ideally, each subsequence $\mathbf{X}_i^j$ would be matched with a specific label to generate accurate incremental prediction values, yet the reality of medical imaging is such that records typically conclude with a singular, final diagnosis for the entire collection of images. For a dataset of even modest size, say $n \approx 1000$, the task of securing expert labels for every subsequence across all sequences becomes daunting, as the number of instances per sequence, $m_i$ usually ranges between 10 and 100. It is also insufficient to directly utilize the sequence-level label as a stand-in for the labels of individual subsequences, as any given subsequence may lack instances that are indicative of positive findings. In the example of Figure 2, it would be incorrect to train the first subsequence $X_i^1$ on the positive label $y_i$ with the same weight as training the last subsequence $X_i^6$, as doing so would result in unrealistic incremental predictions. Thus, in Section 3.2, we propose an innovative modeling and training approach designed to navigate this challenge, enabling the generation of meaningful predictions for subsequences.

### 3.2 THE BiSMIL MODEL AND TRAINING PROCESS

To better capture the sequential nature of clinical imaging, we design a bidirectional transformer BiSMIL and a corresponding novel training algorithm. An overview of the BiSMIL model is shown

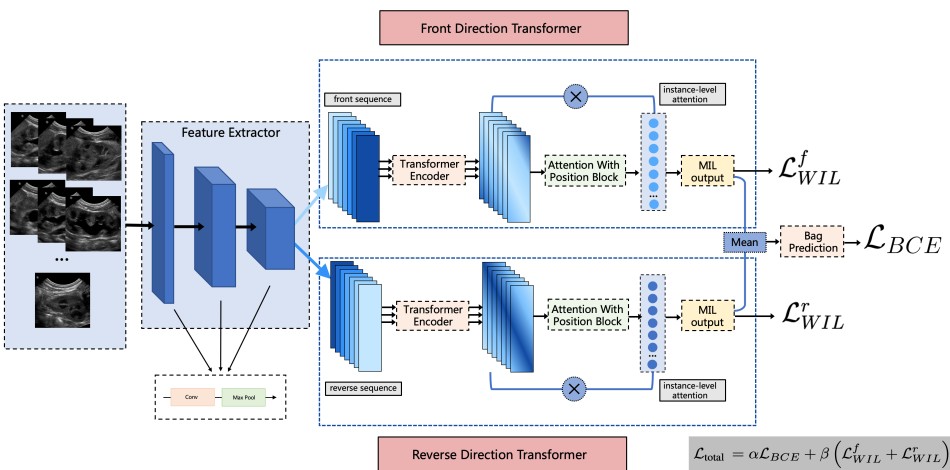

Figure 2: Architecture of the proposed BiSMIL Model.

in Figure 2, and we leave full details to the Appendix. We denote the model as $g(\cdot, \cdot; \boldsymbol{\theta})$ where the inputs represent the front and reverse sequence. We detail some key design decisions below:

**Bidirectional Transformer** We utilize a bidirectional Transformer to effectively combine the raw input features extracted through the convolutional layers. In particular, we consider both "front" and "reverse" directions of the image sequence. This is because scanning direction is usually a preference based on a particular clinician, and therefore we design our model to be robust to sequence reversals.

**Position Encoding in Attention Module** To capture the relative order of instances within a sequence, we augment the attention-based MIL model (Ilse et al., 2018) by combining linear and Gaussian position embeddings into the attention mechanism. The linear embedding captures the sequential order of instances, while the Gaussian embedding is designed to encourage robustness in the reverse sequence. Specifically, the position encoding layer constructs a matrix $\mathbf{P} \in \mathbb{R}^{m_i \times 2}$ for a sequence comprising $m_i$ instances, defined as followed:

$$P_{i,\text{linear}} = \frac{i}{m_i - 1} \tag{1}$$

$$P_{i,\text{gaussian}} = \exp\left(-\frac{(2i - m_i)^2}{2m_i^2}\right) \tag{2}$$

This positional encoding matrix $\mathbf{P}$ is concatenated with the remaining features in the Attention block before the attention function is calculated.

### 3.3 Subsequence-Aware Training Procedure

The goal of the SMIL Framework is to produce incremental predictions that achieve both high final accuracy and high early accuracy, while being faithful to the (unobserved) subsequence labels.

To satisfy all these objectives, we design a novel training procedure for the BiSMIL model. For each dataset, we first determine a minimum subsequence percentage $\gamma \geq 50\%$ so that the minimum subsequence length for training each sequence is $\lfloor \gamma m_i \rfloor$. We can utilize cross-validation to select the optimal $\gamma$ for each dataset, but our experiments suggest $\gamma \in [50\%, 70\%]$ generally produce the best results. We include a sensitivity analysis of the $\gamma$ values on our datasets in Appendix A.3.1 to reflect this fact.

Then for each $l \in \{\lfloor \gamma m_i \rfloor, \cdots, m_i\}$, we take an $l$-length subsequence for both the front and reverse directions. The front direction receives $\{\mathbf{x}_{i1}, \cdots, \mathbf{x}_{il}\}$, and the reverse direction receives

$\{\mathbf{x}_{im_i}, \cdots, \mathbf{x}_{il}\}$. Since $\gamma \geq 50\%$, the union of the two directions covers all samples in the $i$th sequence while each direction only learns from a $l$-sized subsequence. We denote the incremental prediction from the length-$l$ subsequence training as $p_{il}$ while those from the front and reverse directions as $p_{il}^f$ and $p_{il}^r$ respectively.

To setup the loss function, first we consider the final union output $p_{im_i}$. Given that both directions have seen the full sequence, we can evaluate $p_{im_i}$ with the standard BCE loss $\mathcal{L}_{\text{BCE}}$, written as:

$$\mathcal{L}_{\text{BCE}} = -\frac{1}{n} \sum_{i=1}^{n} y_i \log(p_{im_i}) + y_i \log(1 - p_{im_i})$$

To encourage learning on the subsequences, we additionally consider evaluating the outputs from the individual directions. Define $m_i^\gamma := m_i - \lfloor \gamma m_i \rfloor + 1 = |\{\lfloor \gamma m_i \rfloor, \cdots, m_i\}|$ as the total number of subsequences evaluated for sequence $i$ under $\gamma$. As noted previously, naively training each subsequence on the sequence-wise label $y_i$ produces distorted results. Therefore, we consider a modified BCE that is weighted over the $m_i^\gamma$ subsequences, where smaller subsequences are weighted less to account for the fact that the smaller subsequences might not yet have seen a key image that could contribute to a successful prediction. We denote this objective as the *weighted incremental loss* ($\mathcal{L}_{\text{WIL}}$), and write for $a \in \{f, r\}$:

$$\mathcal{L}_{\text{WIL}}^a = -\frac{1}{nm_i^\gamma} \sum_{i=1}^{n} \sum_{l=\lfloor \gamma m_i \rfloor}^{m_i} w_{il} \left( y_i \log(p_{il}^a) + y_i \log(1 - p_{il}^a) \right)$$

$$w_{il} = \frac{e^{(l-m_i)/2}}{\sum_{j=\lfloor \gamma m_i \rfloor}^{m_i} e^{(j-m_i)/2}}.$$

Here we utilize softmax weights $w_i$ to strongly penalize longer subsequences and reflect the higher probability that a key image has appeared in the sequence so the prediction should match the bag-level label. Then, the total model loss is a combination of the weighted incremental loss and the BCE loss:

$$\mathcal{L}_{\text{total}} = \alpha \mathcal{L}_{\text{BCE}} + \beta \left( \mathcal{L}_{\text{WIL}}^f + \mathcal{L}_{\text{WIL}}^r \right)$$

$\alpha, \beta$ can be tuned to better suit individual datasets though we have found that $\alpha = \beta = 0.5$ performs well empirically. This hybrid loss function allows the model to balance the objective to optimize for a correct final prediction and a correct sub-sequence prediction. The training procedure is formally recorded in Algorithm 1. For inference on a particular sequence $\mathbf{X}_i$, contrary to the training procedure, we provide $\{\mathbf{x}_{i1}, \cdots, \mathbf{x}_{il}\}$ and $\{\mathbf{x}_{il}, \cdots, \mathbf{x}_{i1}\}$ to the front and reverse directions respectively for each $l$-length subsequence. This ensures that the BiSMIL model is not "looking ahead" when evaluating any sample. The inference procedure is recorded in Algorithm 2.

---

**Algorithm 1** BiSMIL Model Training

1: **Input:** Dataset $\mathbb{D} = (\mathbf{X}_i, y_i)_{i=1}^n$, BiSMIL Model $g(\cdot, \cdot; \boldsymbol{\theta}_0)$ with initialized parameters $\boldsymbol{\theta}_0$, Training epochs $T$, Minimum subsequence percentage $\gamma \in [0.5, 1]$
2: $k \leftarrow 0$
3: **for** $t = 1$ to $T$ **do**
4:     **for** $i = 1$ to $n$ **do**
5:         For each sequence $\mathbf{X}_i$, compute minimum sub-sequence length $\eta = \lceil \gamma m_i \rceil$
6:         **for** $l = \eta$ to $m_i$ **do**
7:             Evaluate $g(\{\mathbf{x}_{i1}, \cdots, \mathbf{x}_{il}\}, \{\mathbf{x}_{im_i}, \cdots, \mathbf{x}_{i,m_i-l}\}; \boldsymbol{\theta}_k)$ to acquire direction-wise predictions $p_{il}^f$ and $p_{il}^r$ and overall prediction $p_{il}$.
8:         **end for**
9:         Compute the aggregate loss $\mathcal{L}_{\text{total}} = \alpha \mathcal{L}_{\text{BCE}} + \beta \left( \mathcal{L}_{\text{WIL}}^f + \mathcal{L}_{\text{WIL}}^r \right)$
10:         Run backward propagation to acquire $\boldsymbol{\theta}_{k+1}$
11:         $k \leftarrow k + 1$
12:     **end for**
13: **end for**
14: **Output:** Trained Model $g(\cdot, \cdot; \boldsymbol{\theta}_k)$

---

---

**Algorithm 2** BiSMIL Model Evaluation

---

1: **Input:** Sequence of Instances $\mathbf{X}_i$, Trained BiSMIL Model $g(\cdot, \cdot; \boldsymbol{\theta}_k)$
2: **for** $l = 1$ to $m_i$ **do**
3:    Evaluate $g(\{\mathbf{x}_{i1}, \cdots, \mathbf{x}_{il}\}, \{\mathbf{x}_{il}, \cdots, \mathbf{x}_{i1}\}; \boldsymbol{\theta}_k)$ to acquire direction-wise predictions $p_{il}^f$ and $p_{il}^r$ and overall prediction $p_{il}$.
4: **end for**
5: **Output:** Incremental Predictions $p_{i1}, \cdots, p_{im_i}$

---

# 4    SMILU: A Sequence-Aware, Interpretable Uncertainty Metric

In many real-world scenarios, beyond accurate predictions, there is a significant need to understand how *certain* a model is in making the prediction. This is particularly critical in the sequential clinical imaging setting where the certainty in the current prediction can help the clinician determine whether to continue, modify or terminate an imaging sequence. To further improve the applicability of the SMIL Framework, we introduce SMILU, a sequence-aware, interpretable uncertainty metric that combines two uncertainty representations to provide clinicians with a useful tool to determine the certainty of a MIL model.

## 4.1    Dispersion and Sequence-Based Uncertainty

The SMILU metric is inspired by the variability observed in incremental predictions across different bags. Intuitively, if a sequence's incremental predictions quickly converge to 0 or 1, the model is more certain about that sequence. Conversely, if the predictions fluctuate significantly, the model is likely to be less certain about the predictions. We consider two key measurements of uncertainty: sequence dispersion uncertainty, and output uncertainty, and combine the two metrics to form our SMILU metric $\mathcal{U}_{\text{SMIL}}$.

**Sequence Dispersion Uncertainty.** Given a set of output probabilities $\mathbf{p}_i = \{p_{ij}\}_{j=1}^{m_i}$ for a sequence of instances, we employ the standard deviation, denoted as $\mathcal{S}$ to capture the dispersion of the sequence.

$$\mathcal{S}(\mathbf{p}_i) = \begin{cases} \sqrt{\frac{1}{m_i-1} \sum_{j=1}^{m_i} (p_{ij} - \bar{p}_i)^2}, & \text{if } n \geq 2 \\ \min(|p-0|, |p-1|), & \text{if } n = 1 \end{cases} \tag{3}$$

Here, $\bar{p}_i = \frac{1}{m_i} \sum_{j=1}^{m_i} p_{ij}$ is the mean output. This metric captures the innate variability of the model output - if the predictions are fluctuating significantly across the sequence, then it is likely that the model is uncertain of its prediction.

**Output Uncertainty.** Another dimension of uncertainty is output uncertainty. For every prediction $p_{ij}$, the output uncertainty can be defined as $p_{ij}(1 - p_{ij})$. Then we take into account that earlier predictions should be accounted less than later predictions, as earlier predictions have likely not yet seen significant information. We again utilize softmax weights to create the final metric $\mathcal{O}$:

$$\mathcal{O}(\boldsymbol{p}_i) = \frac{\sum_{j=1}^{m_i} s_{ij} \cdot |p_{i,j+1} - p_{ij}|}{\sum_{i=1}^{n-1} s_i}, \tag{4}$$

$$s_{ij} = \frac{e^{(j-m_i)/2}}{\sum_{l=1}^{m_i} e^{(l-m_i)/2}}, \quad i = 1, 2, ..., m_i \tag{5}$$

We then propose a weighted average of the two uncertainty components to form the SMILU metric:

$$\mathcal{U}_{\text{SMIL}} = \mathcal{S} \times w_s + \mathcal{O} \times w_o \tag{6}$$

The weights can vary depending on the particular application. We demonstrate the effectiveness of the SMILU metric in Section 5.3.

# 5    Experiments

In this section, we conduct extensive experiments across three datasets to validate the efficacy of our proposed model and the accompanying uncertainty metric. Our results demonstrate: (i) state-of-the-art performance by the BiSMIL model for both final prediction and subsequence prediction and (ii)

| Model | Dataset | Accuracy | Precision | Recall | F1 Score |
|---|---|---|---|---|---|
| SA-DMIL (Wu et al., 2023) | UTD Ultrasound | **93.1 ± 1.8** | **95.3 ± 1.4** | 90.0 ± 0.7 | 92.6 ± 1.1 |
| | RSNA | 76.1 ± 0.9 | 79.2 ± 0.5 | 62.3 ± 0.9 | 69.7 ± 1.0 |
| | CoV-2 CT | 71.9 ± 1.4 | 81.4 ± 1.3 | 82.5 ± 1.1 | 80.6 ± 0.7 |
| MaxPool (Wang et al., 2019b) | UTD Ultrasound | 91.5 ± 0.4 | 94.5 ± 0.6 | 86.7 ± 0.8 | 92.0 ± 0.7 |
| | RSNA | 71.3 ± 1.0 | 69.1 ± 1.3 | 60.2 ± 2.1 | 64.3 ± 1.5 |
| | CoV-2 CT | 74.3 ± 1.1 | 77.9 ± 0.4 | 91.3 ± 0.9 | 84.6 ± 0.5 |
| ADMIL (Ilse et al., 2018) | UTD Ultrasound | 92.2 ± 0.8 | 93.4 ± 1.7 | 89.0 ± 1.4 | 91.6 ± 1.1 |
| | RSNA | 71.2 ± 1.2 | 68.2 ± 0.7 | 61.0 ± 1.3 | 64.0 ± 1.6 |
| | CoV-2 CT | 75.7 ± 1.3 | 77.7 ± 1.5 | **95.6 ± 0.7** | 85.7 ± 0.3 |
| SiSMIL | UTD Ultrasound | **93.3 ± 1.9** | **96.5 ± 1.2** | **91.8 ± 0.8** | **94.0 ± 1.5** |
| | RSNA | **78.0 ± 1.4** | **82.6 ± 0.9** | 60.5 ± 0.8 | 69.8 ± 1.1 |
| | CoV-2 CT | 76.7 ± 1.6 | **85.4 ± 1.0** | 86.9 ± 0.9 | 84.6 ± 1.2 |
| BiSMIL | UTD Ultrasound | **94.2 ± 0.7** | **97.2 ± 0.9** | **92.3 ± 1.2** | **94.5 ± 0.6** |
| | RSNA | **80.4 ± 2.1** | **81.1 ± 1.0** | **66.8 ± 0.8** | **73.1 ± 1.4** |
| | CoV-2 CT | **80.0 ± 1.2** | **86.5 ± 1.1** | 88.7 ± 1.1 | **87.0 ± 0.9** |

Table 1: Accuracy, Precision, Recall, F1 score of BiSMIL, SiSMIL and comparison models across the UTD, RSNA, and COV-2 CT dataset, averaged over 5 independent trials. We also showcase the standard deviations of these metrics. For each metric, the best-performing model, along with models that have statistically indistinguishable performance at the 95% level are highlighted.

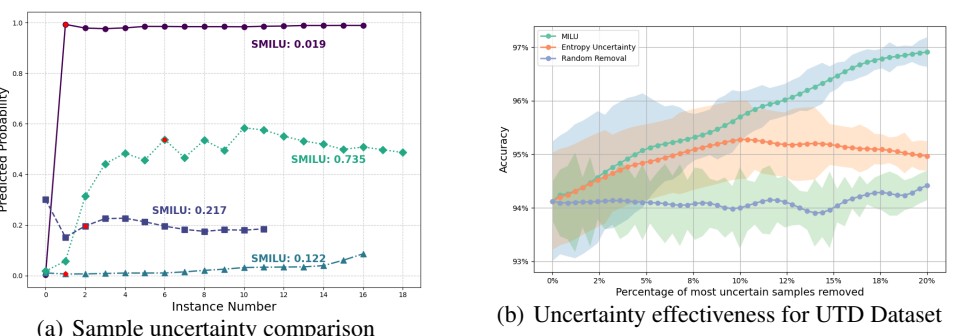

(a) Sample uncertainty comparison

(b) Uncertainty effectiveness for UTD Dataset

Figure 3: (a) Incremental predictions of selected samples on the UTD dataset and their corresponding SMILU uncertainty metric. The red dot indicates the image with the highest attention score. (b) Accuracy of the BiSMIL model on the UTD dataset as samples top-ranked in various uncertainty metrics are removed. The shaded area represents the 95% confidence band.

efficacy of the SMILU uncertainty metric. We further provide an open-source implementation of our framework at our github repository.

We first introduce our real-world datasets. For all of our experiments, we designated 70% of the data for training, 20% for testing, and the remaining 10% for validation. The detailed experimental setup is in the Appendix.

**UTD Classification Dataset:** Urinary tract dilation (UTD) is a relatively common medical condition in children that affects approximately $1 - 2\%$ of the infant population in the United States (Chow et al., 2017; Nguyen et al., 2022). UTD is generally detected through ultrasound, and graded from P1 to P3 in order of increasing severity. We evaluate our algorithm on a novel UTD classification dataset, acquired with IRB approval, that consists of data from 1,184 patients each with multiple ultrasound scans forming a sequence. The average number of scans across each patient is 11.7. We collapse the different grades of UTD to a binary label of $\{0, 1\}$ that indicates if UTD is present in the sequence of ultrasound scans. In the overall dataset, the prevalence of UTD is 48.3%.

**RSNA Dataset:** This dataset is obtained from the 2019 Radiological Society of North America (RSNA) challenge. We randomly selected a subset from the entire RSNA Dataset, comprising 50,862 brain CT slices across 1,175 patients. Following the preprocessing protocol established

in Wu et al. (2021), each CT slice was subjected to three distinct window settings applied to the original Hounsfield Units. This process models after standard radiologist practice, which adjusts the window Width (W) and Center (C) to enhance the visualization of specific tissues in brain CTs. The chosen settings were brain (W: 80, C:40), subdural (W:200, C:80), and soft tissue (W:380, C: 40). Subsequently, all images were resized to a uniform dimension of $224 \times 224$ pixels and normalized within the range [0, 1]. In the original dataset, there are five types of brain hemorrhage, and we create the sequence-level binary label where a positive label indicates if any of the five types of hemorrhage is present. In total, 41.7% of patients were labelled positive.

**SARS-CoV-2 CT-Scan Dataset:** The SARS-CoV-2 CT-Scan dataset incorporates 4,173 CT scans from 210 unique patients (Soares et al., 2023). The dataset contains 80 (38%) COVID-19 positive patients, along with 80 (38%) patients that exhibit other pulmonary conditions. For the purpose of the experiment, we utilized a sequence-level binary label where positive indicates the patient has at least one pulmonary condition.

We compare the performance of our BiSMIL model against the leading benchmark of SA-DMIL (Wu et al., 2023) [1], and commonly used MIL models such as MaxPool (Wang et al., 2019b) and ADMIL (Ilse et al., 2018). We also provide a comparison to a one-directional variant of our BiSMIL model where we remove the reverse direction, denoted as the SiSMIL model in the following experiments.

## 5.1 FINAL PREDICTION ACCURACY

We first compare BiSMIL against benchmarks in final prediction accuracy, where the full sequence is provided to all algorithms. To ensure fairness in comparison, all results are based on the best hyperparameter settings as reported in the original publications. In Table 1, we record the Accuracy, Precision, Recall, and F1 Score of all models across the three medical imaging datasets. We observe that across all metrics and all datasets, the BiSMIL model outperforms all leading benchmarks, often with statistical significance. These results reflect the importance of leveraging sequential information in clinical imaging datasets. In Appendix A.3.2, we demonstrate that the position embedding module is an important driver of the BiSMIL's performance, providing further evidence of the importance of the image ordering. Furthermore, we observe that the BiSMIL model achieves moderate, but statistically significant gains compared to the SiSMIL model, which suggests that bidirectionality provides extra information that can improve the effectiveness of the model.

## 5.2 SUBSEQUENCE PREDICTION ACCURACY

To further understand the performance of our BiSMIL model, we compare the accuracy of the BiSMIL model against the three comparison models when only a subsequence of instances are revealed. We only include the UTD and RSNA datasets for this experiment as the COVID CT scan dataset is insufficiently large to draw conclusions. We observe in Figure 5.2 that in general, as more instances are added, the performance of all models increase. However, we observe that the BiSMIL model achieves high prediction accuracy significantly earlier than comparing methods: for the UTD dataset, with just 50% of the instances the BiSMIL model achieves an accuracy that is comparable to ADMIL with 100% of the instances and SA-DMIL with 70% of the instances. Alternatively, this means that BiSMIL can achieve the same accuracy with $30 - 50\%$ fewer instances compared to benchmarks. The results are generally similar with the RSNA dataset. These results, together with Table 1, demonstrate that our novel training procedure and bidirectional architecture can simultaneously achieve high final accuracy while providing exceptional early accuracy.

## 5.3 EFFECTIVENESS OF SMILU

We further present the value of sequence information by demonstrating the effectiveness of the sequence-aware uncertainty metric, SMILU. Figure 3 (a) illustrates the sequence of incremental predictions for a few samples from the UTD dataset, and the resulting SMILU metric. We observe that instances with more fluctuation and slower convergence exhibit higher SMILU scores. Samples

---

[1]Wu et al. (2023) did not specify the exact random subset of the selected RSNA Dataset and therefore our results of SA-DMIL differ from the exact results reported in Wu et al. (2023). We sampled 5 random subsets from the RSNA dataset and confirmed that our subset results are representative. Such results are included in Appendix A.3.3.

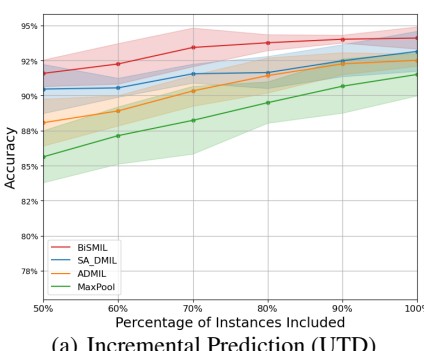 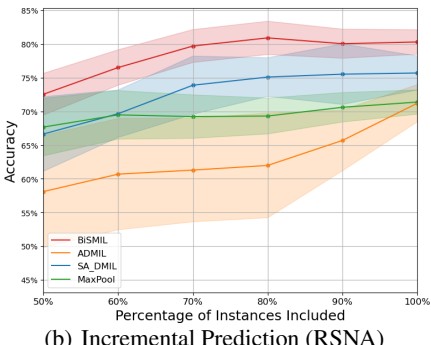

(a) Incremental Prediction (UTD)     (b) Incremental Prediction (RSNA)

Figure 4: Comparison of BiSMIL with benchmark methods on the accuracy of incremental predictions. The shaded area represents the 95% confidence band.

with significant fluctuations are often challenging to classify, as it indicates a mix of weak positive and negative signals. To provide evidence that the SMILU metric can capture the most challenging cases to classify, in Figure 3 (b), we plot the accuracy of the BiSMIL model on the UTD dataset when we remove the top-ranked samples in the SMILU metric. We compare the accuracy trend with removing top-ranked samples in entropy, a common uncertainty metric that depends only on the final output. We observe that removing 20% of the most uncertain predictions using the SMILU metric improved the accuracy more significantly compared to entropy or random removal. This demonstrates that the SMILU metric can capture difficult-to-predict instances better than classic metrics based purely on the final output.

## 6 LIMITATIONS

Despite the promising results achieved by the BiSMIL model and the SMILU uncertainty metric across various datasets, this study includes multiple limitations.

First, we focus only on binary classification, while many medical imaging tasks admit natural multi-class classification or regression formulations. It remains to be seen if a similar approach can also perform in these contexts. Second, the proposed SMILU metric provides a novel approach to quantify uncertainty in sequence predictions but the validation of this metric is primarily empirical. A theoretically-grounded metric or formulation could further improve the usability of the metric for the clinical decision process. Finally, although our experiments encompass a range of conditions, the generalizability of our model to other tasks, such as MRI and X-ray, are untested. External validation of datasets from different institutions would also enhance the robustness of the performance.

## 7 CONCLUSION

In conclusion, our research introduces the Sequential MIL (SMIL) framework that systematically incorporates the sequential nature of clinical imaging into the MIL framework. The SMIL framework presents new tradeoffs and challenges for MIL methods, as it is important in the SMIL framework to provide accurate, early incremental predictions. We propose a bidirectional Transformer model, BiSMIL, along with a novel training procedure that aims to balance the importance of an accurate final prediction and an accurate early prediction. Experiments on multiple medical image datasets demonstrate that the BiSMIL model is able to outperform current benchmarks on final prediction accuracy while significantly improving the accuracy of incremental predictions. We further propose an interpretable, sequence-aware uncertainty metric, SMILU, that is able to better capture difficult-to-predict instances compared to metrics that rely solely on the final output. This again demonstrates the importance of incorporating the sequential nature of the setting.

Although this work has largely focused on clinical imaging settings, there are other important settings that share this sequential multi-instance learning structure. Common examples include time-series

event prediction and online video analysis. We hope this work can encourage further method development within this setting.

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

# A  APPENDIX

## A.1  DETAILS OF EXPERIMENT SETTING

For each dataset, we designated 70% of the data for training, 20% for testing, and the remaining 10% for validation. Every image is resized to a uniform dimension of $224 \times 224$ in the experiments.

The hyper-parameter settings for different models are shown in Table 2 and 3. For SA-DMIL, we employ the best hyperparameters shown in the original paper Wu et al. (2023).

| Hyper-parameters | Values |
|---|---|
| epoch number | 60 |
| batch size | 1 |
| learning rate for SiSMIL & BiSMIL | $2e^{-5}$ |
| optimizer | Adam |
| weight decay rate | $1e^{-4}$ |
| dropout | 0.2 |
| number of transformer layers | 2 |
| number of head | 8 |
| feedforward network dimension | 128 |
| clip ratio | 0.5-0.7 |
| $\beta$ for weighted incremental loss | 0.5 |

| Hyper-parameters | Values |
|---|---|
| epoch number | 40 |
| batch size | 1 |
| learning rate for ADMIL & MaxPool | $1e^{-4}$ |
| optimizer | Adam |
| weight decay rate | $1e^{-4}$ |

Table 2: Hyperparameters for SiSMIL & BiS-MIL

Table 3: Hyperparameters for ADMIL & MaxPool

## A.2  DETAILS OF BISMIL

As shown in Figure 2, the BiSMIL model consists of a feature extractor, two transformer encoder blocks and also two attention modules with position encoding.

**Feature Extractor** The Feature Extractor utilizes a VGG backbone composed of convolutional layers, batch normalization, and max pooling layers. We employ six such blocks to extract features from the input bags.

**Transformer Encoder Block** In the Transformer block, following the classic structure, we assume the input front subsequence is: $\mathbf{X}_i = \{\boldsymbol{x}_{i,1}, \boldsymbol{x}_{i,2}, \ldots, \boldsymbol{x}_{i,n}\}$. Then, within the Transformer block, we perform:

$$\mathbf{Q}^\ell = \mathbf{X}_i^{\ell-1}\mathbf{W}_Q, \quad \mathbf{K}^\ell = \mathbf{X}_i^{\ell-1}\mathbf{W}_K, \quad \mathbf{V}^\ell = \mathbf{X}_i^{\ell-1}\mathbf{W}_V, \quad \ell = 1\ldots L$$

$$\text{head} = \text{SA}(\mathbf{Q}^\ell, \mathbf{K}^\ell, \mathbf{V}^\ell) = \text{softmax}\left(\frac{\mathbf{Q}^\ell(\mathbf{K}^\ell)^T}{\sqrt{d_q}}\right)\mathbf{V}^\ell, \quad \ell = 1\ldots L$$

$$\text{MSA}(\mathbf{Q}^\ell, \mathbf{K}^\ell, \mathbf{V}^\ell) = \text{Concat}(\textbf{head}_1, \text{head}_2, \ldots, \text{head}_h)\mathbf{W}^O, \quad \ell = 1\ldots L$$

$$\mathbf{X}_i^\ell = \text{MSA}(\text{LN}(\mathbf{X}_i^{\ell-1})) + \mathbf{X}_i^{\ell-1}, \qquad \ell = 1\ldots L$$

where $\mathbf{W}_Q \in \mathbb{R}^{d \times d_q}, \mathbf{W}_K \in \mathbb{R}^{d \times d_k}, \mathbf{W}_V \in \mathbb{R}^{d \times d_v}, \mathbf{W}_O \in \mathbb{R}^{hd_v \times d}$, head $\in \mathbb{R}^{(n+1) \times d_v}$, SA denotes Self-Attention layer, $L$ is the number of MSA block, $h$ is the number of heads in each MSA block, and Layer Normalization (LN) is applied before every MSA block (Vaswani et al., 2017).

**Attention Module with Position Encoding** After all features pass through the Transformer encoder block, the front (reverse) subsequences enter the attention module with position encoding. The features are concatenated with position encoding before each pass through a linear layer and Tanh function. Each time, they are concatenated with:

$$P_{i,\text{linear}} = \frac{i}{m_i - 1}$$

$$P_{i,\text{gaussian}} = \exp\left(-\frac{(2i - m_i)^2}{2m_i^2}\right)$$

Finally, the attention values are obtained through the Softmax function. These are weighted averaged with Transformer features and sent to a simple classifier consisting of a linear layer and activation function to produce the output probability.

## A.3 ABLATION STUDY

### A.3.1 ABLATION STUDY OF CLIP RATIO $\gamma$ ON DIFFERENT DATASETS

We present an ablation study of the clip ratio $\gamma$ for different dataset. The results, detailed in the table below, represent averages derived from five independent random seed experiments.

Table 4: Performance metrics for different datasets at various $\gamma$ levels

|  | UTD | | | | RSNA | | | | Covid | | | |
|---|---|---|---|---|---|---|---|---|---|---|---|---|
| $\gamma$ | Acc | Pre | Rec | F1 | Acc | Pre | Rec | F1 | Acc | Pre | Rec | F1 |
| $\gamma = 0.5$ | 93.2 | 95.5 | 91.2 | 93.4 | 78.9 | 79.9 | 63.7 | 70.7 | 75.2 | 84.6 | 85.0 | 83.7 |
| $\gamma = 0.6$ | 93.7 | 95.9 | **92.1** | 94.0 | **80.4** | **81.1** | **66.8** | **73.1** | **80.0** | 86.5 | 88.7 | **87.0** |
| $\gamma = 0.7$ | **94.2** | **97.2** | **92.3** | **94.5** | 78.0 | 78.9 | 61.4 | 69.1 | 78.6 | **88.7** | 84.4 | 85.3 |
| $\gamma = 0.8$ | 93.4 | 95.6 | 91.8 | 93.7 | 78.7 | 79.6 | 65.6 | 71.6 | 72.9 | 85.0 | 80.0 | 81.3 |
| $\gamma = 0.9$ | 92.8 | 96.2 | 89.8 | 93.0 | 78.4 | **80.9** | 63.2 | 71.0 | 78.1 | 82.1 | **92.5** | 86.4 |
| $\gamma = 1.0$ | 93.3 | 95.9 | 91.3 | 93.5 | 76.9 | 80.5 | 61.0 | 69.4 | 77.6 | 84.1 | 88.1 | 85.6 |

### A.3.2 ABLATION STUDY OF BISMIL POSITION EMBEDDING

We present an ablation study on the position embedding module of the BiSMIL module, which indicates that the position embedding significantly contributes to the accuracy of the model and suggests that knowledge of the relative order of the features is indeed useful for understanding the images.

Table 5: Ablation Study of Position Embedding Module for BiSMIL on Different Datasets

| | UTD | | | | RSNA | | | | Covid | | | |
|---|---|---|---|---|---|---|---|---|---|---|---|---|
| Position Embedding? | Acc | Pre | Rec | F1 | Acc | Pre | Rec | F1 | Acc | Pre | Rec | F1 |
| Yes | **94.2** | **97.2** | **92.3** | **94.5** | **80.4** | **81.1** | **66.8** | **73.1** | **80.0** | **86.5** | 88.7 | **87.0** |
| No | 92.0 | 95.0 | 89.9 | 92.4 | **79.9** | 79.8 | **66.1** | **72.3** | 76.7 | 82.7 | **88.7** | 85.1 |

### A.3.3 SENSITIVITY ANALYSIS OF RSNA SUBSET SELECTION

Table 6: SA-DMIL Results across Different Subsets of RSNA

| Subset | Acc | Pre | Rec | F1 |
|---|---|---|---|---|
| 1 | 76.1 | 79.2 | 62.3 | 69.7 |
| 2 | 75.7 | 76.8 | 64.5 | 70.1 |
| 3 | 74.5 | 77.2 | 66.8 | 71.6 |
| 4 | 78.3 | 79.8 | 72.3 | 75.8 |
| 5 | 74.9 | 76.4 | 65.8 | 70.7 |

