# OpenReview forum: "Towards Interpretable, Sequential Multiple Instance Learning: An Application to Clinical Imaging"
_ICLR.cc/2025/Conference — Submitted to ICLR 2025_

### Official Review · Reviewer_Vuz2 · 2024-10-18

**Soundness:** 2
**Presentation:** 2
**Contribution:** 1
**Rating:** 3
**Confidence:** 3

**Summary:**

This paper presents a method for medical image classification, BiSMIL, which adopts a sequential multiple instance learning (MIL) framework that considers the “order” of instances. The proposed method outperforms other MIL baselines on three medical imaging benchmark datasets, and the authors put forth an uncertainty metric that assesses variability in instance-wise predictions.

**Strengths:**

- The organization and layout of the paper is mostly sound.
- Experiments demonstrate strong performance over multiple datasets and baselines.
- The formulation of CT and ultrasound image analysis as a sequential MIL problem is unique to me.

**Weaknesses:**

- The setting and motivation for adopting a “sequential” framework to medical image analysis is completely unclear. Every imaging modality and use case is different, so I do not know what the authors mean by phrases like “the sequential nature of clinical imaging.” Provide a concrete example (perhaps with a figure) of a medical imaging use case that is sequential purely for illustration purposes. It is unclear whether this refers to longitudinal imaging (multiple images acquired over potentially years or months to track disease progression), medical videos (multiple images over a short time period forming a video), or something else.
- The authors do not convey understanding of the application domain. I don’t follow why “CT scan levels” are any more or less “ordered” than image patches of a large histopathology slide. Justify this dichotomy that the authors set up. Comments such as “clinicians are creating the images sequentially” and “scanning direction is usually a preference based on a particular clinician” display a misunderstanding of clinical workflows; it is not usually accurate to say that clinicians are “creating” images, and I am not aware of a medical image interpretation use case where a clinician will assess a “sequence” of images in the forward and reverse direction.
- Description of datasets used is insufficient and causes further confusion, where it could be used to clarify the above points. E.g., it is unclear whether the ultrasound dataset is image- or video-based; whether the “sequence” is formed from multiple images/videos in the same study vs. multiple studies from the same patient; etc. See specific questions below.
- If I am understanding the datasets correctly (which I am not sure that I am), then I fail to see why only a select few MIL methods are used as comparative baselines. If the authors are performing classification from 3D CT volumes, there are a wealth of methods for volumetric medical image analysis to pull from beyond MIL approaches.

**Questions:**

- What do the authors mean by “clinicians are creating the images sequentially to discover features of interest”? Sequentially over *time*? What time-scale (e.g., frames in a video or scans captured over many years)? Perhaps a figure visualizing one example use case could clarify this.
- In what sense are “CT scan levels” and “ultrasound” sequential? In what sense does order of “instances” (however you define this) matter? Expand on this. This is not a perspective I am familiar with, so it warrants further explanation.
- Provide more details about the nature of the datasets used. Does the UTD dataset consist of ultrasound images or videos? What exactly is the “sequence” in this setting: frames in a video? videos from different “views” in a study? different studies over a patient’s longer-term trajectory? For CT datasets, do the individual 2D slices in the 3D volume form the “sequence”?

**Minor comments/questions:**
- Ideally, provide citations or examples of the “surge of availability of imaging”
- I would merge the first two paragraphs – awkward that the first paragraph is two sentences.
- Awkward wording: “each image study often contains multiple number of image instances” -> “Each medical imaging study often contains multiple image acquisitions (“instances”) perhaps
- “clinicians are creating the images sequentially”. I would not say clinicians are “creating” images (perhaps “acquiring”?), and I do not know what is meant by “sequentially” – there are many ways to interpret this word.
- Change “patient radiation exposure” -> “, in some cases, patient radiation exposure”. One of the examples given, ultrasound, does not use ionizing radiation
- What is being predicted in Figure 1? Mention briefly in the caption.
- The definition of “sequence” is unclear to me in the context of ultrasound, but let’s presume the authors are considering a sequence to be each of a standard few views that are acquired in a recommended/standard order. Why does this order matter for diagnosis? I am not aware that it does. In applications of ultrasound such as echocardiography, views may be acquired in a recommended order, but the analysis of views simply depends on the task being performed and does not depend on order – often only one or two views are relevant for a particular task.
- For the CT datasets, do I understand correctly that the authors are considering a “sequence” of 2D slices in the 3D volume? If so, why not compare to standard 3D medical image analysis methods?
- What statistical test is performed to compare model performance?

---

> ### Author Response · Authors · 2024-11-25
>
> We thank the reviewer for their detailed feedback. Below, we address each concern and question raised point by point.
>
> **1. Unclear Motivation and Definition of “Sequential Nature”**
>
> We appreciate the reviewer’s comment and have addressed related concerns in Reviewer V1jg’s Point 1. In our work, a “sequence” refers to the ordered set of images collected during a single visit or imaging session. For example, in CT scans, slices are typically acquired following a fixed anatomical direction (e.g., cranial-to-caudal). These anatomical orders inherently contain sequential information, which reflects spatial or contextual relationships crucial for diagnosis.
>
> Traditional MIL methods, which treat instances as unordered, are unable to capture this sequential information, limiting their ability to fully utilize the diagnostic potential of such imaging data. Our framework addresses this gap by explicitly incorporating the order of instances into the model, allowing it to make better use of sequential information between slices. As for the example, during an ultrasound session, a sonographer incrementally captures images along specific anatomical planes, such as longitudinal or transverse planes. You can get a more intuitive understanding of this concept from the example illustrated in Figure 1.
>
> **2. Dichotomy Between CT and Histopathology Image Patches**
>
> Thank you for raising this point. While MIL is indeed a commonly used method for analyzing histopathology slides ([Wang, Jun, et al.](https://arxiv.org/abs/2408.09476)), there is a fundamental difference in how MIL is applied to histopathology slides versus clinical scans. Histopathology slides lack an inherent order; the primary goal is to locate regions within the slide that exhibit disease-specific features, without considering the spatial sequence of patches.
>
> In contrast, clinical scans, such as CT imaging, are typically acquired in a specific direction, such as along the transverse plane or longitudinal plane. This introduces an order between slices, resulting in sequential information. For example, adjacent slices are more similar to one another, while changes observed in different regions along the sequence often provide critical diagnostic insights.
> Leveraging the sequential information inherent in clinical scans can significantly improve model performance. This insight forms one of the key contributions and motivations of our work.
>
>
> **3. Dataset Details**
>
> Thank you for raising this question. The datasets we used are image-based, and the order of the images is arranged according to the sequence numbers from the original DICOM files. This ordering reflects the natural anatomical progression captured during the scan. We believe this sequential structure is evident in both of the publicly available datasets we utilized.
>
> **4. Why Not Compare to 3D Medical Image Analysis Methods?**
>
> We have addressed a related question in our response to **Reviewer V1jg**. We selected MIL methods as baselines because our task operates in a weakly supervised setting, where only bag-level labels are available. In contrast, non-MIL methods, such as 3D convolutional networks, typically require instance-level labels or fixed-size inputs. These requirements make them unsuitable for our problem setting and significantly more challenging to implement in practice due to the difficulty of obtaining such detailed annotations.

---

> ### Author Response · Authors · 2024-11-25
>
> **Responses to Questions**
>
>
> 1. We believe this question is addressed in our response to Weakness 1. A “sequence” refers to the ordered set of images collected during a single visit or imaging session. Figure 1 serves as a good example to illustrate this concept.
>
> 2. As mentioned in our response to Weakness 2, clinical scans, such as CT imaging, are typically acquired in a specific direction, such as along the transverse or longitudinal plane. This introduces an inherent order between slices, resulting in sequential information. For instance, adjacent slices are more similar to one another, while changes observed in different regions along the sequence often provide critical diagnostic insights. This is why order and sequential information are important for clinical scans.
>
> 3. This point is also addressed in our explanation of the datasets. The datasets we used are image-based, and the order of the images is arranged according to the sequence numbers from the original DICOM files, preserving the natural anatomical progression of the scans.
>
> **Responses to Minor Comments**
>
> • We thank the reviewer for their suggestions under the minor comments section, and we will incorporate them in the revised version of the paper.
>
> • Figure 1 represents a partial prediction from a patient sample in the UTD dataset. The increasing probability approaching 1 indicates that the model predicts a high likelihood of this patient being positive for UTD.
>
> • We have already addressed the importance of sequence information and the comparison with other models in our previous responses.
>
> • For statistical significance, we performed t-tests for our experimental results compared to BiSMIL. The results suggest that our findings are statistically significant at the 5% level. We will include a detailed table of these results in the final version.
>
>
> Wang, Jun, et al. "Advances in Multiple Instance Learning for Whole Slide Image Analysis: Techniques, Challenges, and Future Directions." arXiv preprint arXiv:2408.09476 (2024).

---

> ### Comment · Reviewer_Vuz2 · 2024-11-25
>
> **1. Unclear Motivation and Definition of “Sequential Nature”**
>
> I appreciate the explanation, but this does not address my specific concern regarding the utility of the *sequential order of image acquisition in ultrasound*. Why does order matter? To give a concrete example, a common way to measure pediatric ejection fraction is through cardiac ultrasound imaging using two major views: apical 4 chamber (A4C) and parasternal short axis (PSAX). I have no reason to believe that giving a clinician (or deep learning model) the A4C view first followed by PSAX should yield different results than PSAX first followed by A4C.
>
> To reiterate one of my original questions:
> "The definition of “sequence” is unclear to me in the context of ultrasound, but let’s presume the authors are considering a sequence to be each of a standard few views that are acquired in a recommended/standard order. Why does this order matter for diagnosis? I am not aware that it does. In applications of ultrasound such as echocardiography, views may be acquired in a recommended order, but the analysis of views simply depends on the task being performed and does not depend on order – often only one or two views are relevant for a particular task."
>
> **2. Dichotomy Between CT and Histopathology Image Patches**
>
> I agree that modalities like CT are acquired in a "sequential" manner where adjacent slices are similar. However, I feel one could argue that the same is true to some extent for histopathology imaging -- this still feels to me like a false dichotomy. If adjacent CT slices capture similar anatomy, don't adjacent "patches" of a histopathology slide capture similar/nearby aspects of the cells being visualized?
>
> **4. Why Not Compare to 3D Medical Image Analysis Methods?**
>
> I am unsure what is meant specifically by the following: "In contrast, non-MIL methods, such as 3D convolutional networks, typically require instance-level labels or fixed-size inputs." Perhaps I am misunderstanding the nature of the CT dataset -- are labels provided at the slice or scan level? To give an example of my original suggestion: if labels are provided at the slice level, then a scan-level label can be derived, in which case one could train a 3D CNN on whole volumes optimized toward this scan-level label. This would serve as a simple baseline to justify whether the effort of a customized MIL approach is "worth it" in terms of downstream performance.
>
> This paper poses a unique perspective of and approach to medical image analysis that I feel remains opaque and misunderstands clinical reality. Sweeping statements to the effect of "sequential information [is] important for clinical scans" do not help to clarify this perspective. Every type of "clinical scan" is different, particularly the two modalities presented in this paper (CT and ultrasound), for reasons outlined above.

---

> > ### Author Response · Authors · 2024-12-03
> >
> > **1. Unclear Motivation and Definition of “Sequential Nature”**
> >
> > We appreciate the reviewer’s detailed follow-up and the specific example provided. We agree that in some tasks, such as echocardiography, the specific order of views may not significantly impact diagnostic outcomes. However, our framework is designed to address scenarios where the incremental acquisition of images or the scanning process inherently follows a specific directional sequence. For instance, in modalities like CT and some ultrasound scanning, images are often captured along a defined anatomical plane (e.g., cranial-to-caudal or left-to-right), where sequential information provides critical spatial or diagnostic context.
> >
> >
> > **2. Dichotomy Between CT and Histopathology Image Patches**
> >
> > We understand your concern and appreciate the comparison. While adjacent histopathology patches may capture similar cellular regions, these relationships lack the anatomical or spatial continuity inherent in CT scans, where sequential slices reflect real-world anatomy. This anatomical structure in CT sequences allows models to leverage spatial progression, which is often not present in histopathology slides due to their random sampling or stitching processes.
> >
> > **4. Why Not Compare to 3D Medical Image Analysis Methods?**
> >
> > We acknowledge the importance of comparing against 3D CNNs. However, our CT datasets are weakly labeled at the scan (bag) level, without slice-level labels needed to derive instance-level information for 3D CNNs. MIL approaches are specifically suited for such weakly supervised settings. While implementing a scan-level 3D CNN baseline could provide additional insights, we emphasize that this approach would require annotations not available in our datasets. We will include this point and propose comparisons in future work.
> >
> > We appreciate your feedback and agree that the importance of sequential information differs across imaging modalities. We will revise the manuscript to better clarify the context-dependent utility of our approach, avoiding broad generalizations about sequential importance and tailoring our claims to specific use cases presented in this work.

---

### Official Review · Reviewer_EMs4 · 2024-11-02

**Soundness:** 2
**Presentation:** 2
**Contribution:** 2
**Rating:** 5
**Confidence:** 4

**Summary:**

The paper proposes a Sequential Multiple-Instance Learning (SMIL) method, employing a bi-directional Transformer architecture to capture information in both forward and backward directions. Additionally, a sequence-aware uncertainty metric, SMILU, is introduced to quantify the uncertainty in model predictions.

**Strengths:**

At a high level, the paper addresses an important question: how to enhance clinical efficiency in capturing patient images.

**Weaknesses:**

1. Motivation of the study:

1.1 The paper justifies the development of SMIL by stating, "In many clinical imaging settings, clinicians create images sequentially to discover features of interest and often have control over the number of sequential images captured." However, it would strengthen the motivation if a specific clinical setting were described to show where this statement holds true and where the proposed method could be beneficial, since for many clinical protocols are predefined, with standardized imaging procedures. Applying the proposed method in such settings could be challenging, as it would require modifications to existing protocols.

1.2. Similar to above, the paper motivate the value of the proposed method by saying "as the clinician creates more images, the resulting diagnostic accuracy is likely to increase, but it comes at the expense of efficiency and patient radiation exposure." However, not all medical imaging modalities, such as ultrasound, involve radiation exposure. Moreover, it would be valuable to clarify the potential practical efficiency gains that reduce only a few images brings.

2. Method Novelty

The use of a bidirectional architecture for modeling sequence data is a well-established technique. For example, bi-directional models, such as Bi-LSTMs, have long been employed in various contexts to model sequential information.

3. The model is not really doing ``incremental'' prediction. In the inference setting, when the clinician collected a sequence m_1 = {x_1, ..., x_n}, the model generates a prediction p1, when the clinician capture one more image m_2 = {x_1, .., x_n, x_(n+1)}, the model basically treat this as brand new sequence and evaluate again to generate p2. This procedure can be applied with other Multiple-Instance Learning (MIL) methods as well.

**Questions:**

1. It is fine that the author try to relax the classic MIL assumption to incorporate sequence information in real clinical practice. However, it is unclear what specific sequential information the proposed model aims to capture, particularly as the authors also noted that sequence order can sometimes depend on the clinician’s or operator’s preference.

2. In line 194 "the Gaussian embedding is designed to encourage robustness in the reverse sequence". Can you elaborate on this? How does the Gaussian position encoding encourage robustness in the reverse sequence? What about for the forward sequence?

3. It is unclear how the model would inform clinicians about the optimal stopping point for image collection. Given a full sequence (like those experiments in the paper), we can determine the ideal stopping point, but in a real-world setting, where an additional image has not yet been collected, we do not know if additional image would contribute valuable information.

4. As the author noted, training on subsequences with the label of the full sequence may introduce noise. To address this, they use a heuristic that assigns lower weights to shorter subsequences. However, this may still introduce noise. It is unclear how effectively this approach reduces noise, and it would be helpful to understand if this training objective (e.g., removing the second part in line 244) significantly improves model performance.

---

> ### Author Response · Authors · 2024-11-25
>
> We are thankful for the feedback and questions from the reviewer. Here, we would address the individual concerns point-by-point:
>
> **1. Motivation of the Study**
>
> We thank the reviewer for raising this point. The statement in the paper, “In many clinical imaging settings, clinicians create images sequentially to discover features of interest and often have control over the number of sequential images captured,” was intended to highlight that “this sequential nature is currently largely ignored in applications of MIL to clinical imaging.” The primary motivation of our study lies in addressing this gap and demonstrating the value of leveraging sequential information in clinical imaging.
>
> To provide concrete examples:
>
> • In diagnosing urinary tract dilation, a sonographer incrementally captures slices until sufficient diagnostic information is gathered, using the sequential nature of the imaging process to focus on areas of interest.
>
> • In endoscopic imaging, clinicians decide how far to extend the scope based on observed abnormalities, leveraging sequential information to guide decisions interactively.
>
> Regarding the practical value of our method, we have addressed this in our response to Reviewer Azkv, and we provide the relevant portion here for context:
>
>
> The main advantage of achieving high accuracy with fewer instances is that doctors and radiologists can conclude scanning or imaging earlier in the process if they are convinced about the medical diagnosis. For a simplified example, assume that the doctor would conclude the diagnosis is reliable if the accuracy hits above 92%. Then according to Figure 4(a),  the BiSMIL framework would only require the scanning of roughly 60% of the current sequence (compared to 90% for the best comparing method). This would allow radiologists to theoretically improve their workflow efficiency up to 1/60%=1.6x, and be able to scan more patients in a shorter period of time. This has significant real-world implications as currently there is a well-known shortage of radiologists across the globe, including in USA ([Radiology Workforce Shortage in the USA](https://www.acr.org/Practice-Management-Quality-Informatics/ACR-Bulletin/Articles/March-2024/How-Will-We-Solve-Our-Radiology-Workforce-Shortage#:~:text=There%20is%20a%20palpable%20shortage,on%20the%20ACR%20job%20board%20)), Canada ([Health Human Resource Crisis in Canada](https://www.newswire.ca/news-releases/addressing-the-health-human-resource-crisis-in-radiology-departments-across-canada-842727175.html)), and Norway ([Reporting Radiographers in Norway](https://www.sciencedirect.com/science/article/pii/S1078817423000329)). This has led to long wait times ([Wait Times for Imaging Services in Norway.](https://www.ncbi.nlm.nih.gov/pmc/articles/PMC10666297/#:~:text=Conventional%20X%2Dray%20(XR),7.9–11.4%20weeks%20respectively)), which in turn affects negatively patient outcomes ([Impact of Imaging Wait Times on Lung Cancer Prognosis](https://pubmed.ncbi.nlm.nih.gov/24931045/), [Reducing Lung Cancer Diagnosis Time](https://pubmed.ncbi.nlm.nih.gov/17975490/)).
>
> We hope that our sequential framework can help to combat this healthcare crisis by providing strong clinical evidence with fewer images, improving diagnostic efficiency.
>
> **2. Regarding Method Novelty:**
>
> While we acknowledge that position encoding and bidirectional processing are established techniques, the novelty of BiSMIL lies in its ability to systematically integrate sequential information into the MIL framework. To the best of our knowledge, no existing MIL methods effectively incorporate sequential information and can also provide early, accurate predictions. We believe that our work makes a significant and innovative contribution by offering an important insight: for many types of clinical imaging, utilizing the inherent sequential information is crucial for improved performance. Furthermore, we provide a comprehensive framework to address this challenge, including the development of a model architecture, a tailored training method, and an uncertainty metric to enhance interpretability and robustness.
>
> **3. Regarding Incremental Prediction:**
>
> We thank the reviewer for pointing this out, and your understanding is correct—other MIL models can also accept a new subsequence and make predictions. However, because our method incorporates incremental training during the training process, our model is better equipped to utilize the information in $x_{n+1}$ when encountering a new subsequence $m_2$ = \{$x_1, \dots, x_n, x_{n+1}$\} .
>
> This is because our model explicitly learns the sequential relationships between instances, allowing it to effectively leverage the context provided by the prior sequence \{$x_1, \dots, x_n\$} . In contrast, traditional MIL methods, which treat instances as unordered, are unable to capture or utilize this sequential dependency.

---

> ### Author Response · Authors · 2024-11-25
>
> Responses to Questions
>
>
> **1. Sequential Information and Its Incorporation into the Model**
>
> Our approach is motivated by how radiologists typically make decisions when reviewing clinical scans—examining slices one by one and analyzing similarities and changes between adjacent slices to identify disease-specific features of interest. Through the use of position embeddings, incremental training, and a bidirectional architecture, our model is designed to recognize that sequences have an inherent “order.” This allows the model to adjust predictions as new slices are added to the sequence. This capability reflects the sequential information learned by the model, or what we describe as “sequence-awareness.”
>
> **2. Gaussian Embedding and Robustness in Reverse Sequences**
>
> The Gaussian embedding enhances robustness by encoding the relative position of each slice within the sequence. For instance, slices near the edges of the sequence are assigned smaller embedding values, while central slices receive larger values, helping the model understand positional context. Additionally, the symmetry of the Gaussian embedding aligns with the desired invariance in reverse predictions, ensuring that the model produces consistent results regardless of sequence direction.
>
> **3. Optimal Stopping Point for Image Collection**
>
> The practical value of our method lies in improving workflow efficiency by requiring fewer slices to make accurate predictions. For example, with existing datasets (e.g., UTD), assume that a doctor concludes a diagnosis to be reliable if the accuracy exceeds 92%. As shown in Figure 4(a), the BiSMIL framework achieves this threshold using roughly 60% of the sequence, compared to 90% required by the best competing method. This translates to a potential 1.6x improvement in workflow efficiency, allowing radiologists to scan more patients in less time. This has significant real-world implications as currently there is a well-known shortage of radiologists across the globe, including in USA ([Radiology Workforce Shortage in the USA](https://www.acr.org/Practice-Management-Quality-Informatics/ACR-Bulletin/Articles/March-2024/How-Will-We-Solve-Our-Radiology-Workforce-Shortage#:~:text=There%20is%20a%20palpable%20shortage,on%20the%20ACR%20job%20board%20)), Canada ([Health Human Resource Crisis in Canada](https://www.newswire.ca/news-releases/addressing-the-health-human-resource-crisis-in-radiology-departments-across-canada-842727175.html)), and Norway ([Reporting Radiographers in Norway](https://www.sciencedirect.com/science/article/pii/S1078817423000329)). This has led to long wait times ([Wait Times for Imaging Services in Norway.](https://www.ncbi.nlm.nih.gov/pmc/articles/PMC10666297/#:~:text=Conventional%20X%2Dray%20(XR),7.9–11.4%20weeks%20respectively)), which in turn affects negatively patient outcomes ([Impact of Imaging Wait Times on Lung Cancer Prognosis](https://pubmed.ncbi.nlm.nih.gov/24931045/), [Reducing Lung Cancer Diagnosis Time](https://pubmed.ncbi.nlm.nih.gov/17975490/)).
>
> We hope that our sequential framework can help to combat this healthcare crisis by providing strong clinical evidence with fewer images, improving diagnostic efficiency.
>
>
> **4. Noise and Subsequence Training**
>
> We would like to clarify that although the labels of subsequences (sub-bags) could theoretically differ from the whole sequence (bag), our training process mitigates this risk. By using a bidirectional structure and maintaining a minimum subsequence percentage $\gamma \geq 0.5$ , the model always sees the complete whole sequence (albeit split into two complementary subsequences from different directions). This ensures that noise is not introduced.
>
> Additionally, the weighted incremental loss serves two purposes:
>
> (1) helping the two streams learn to recognize sequential patterns in their respective directions, and (2) encouraging the model to make accurate predictions earlier in the sequence. We will consider adding an ablation study of this loss term in the final version to further demonstrate its impact.

---

> ### Comment · Reviewer_EMs4 · 2024-11-30
>
> I would like to thank the authors for providing a detailed response to my concerns and questions. I have read the author's response as well as comments from other reviewer. The author's response has partially addressed my question. Overall i would raise my initial rating.
>
> However, I am still not convinced about the practical value/ how to select optimal stopping point yet (e.g., Discussion on the time/cost saved vs the false negative introduced?).

---

> > ### Author Response · Authors · 2024-12-03
> >
> > We appreciate Reviewer EMs4’s willingness to raise the initial rating and thank you for taking the time to review our detailed responses.
> >
> > Regarding the concern about selecting an optimal stopping point, we acknowledge that this decision would need to be tailored to the specific dataset and clinical context. In practice, this involves a balance between the performance of the model on reduced data and the radiologists’ threshold for acceptable accuracy.
> >
> > For instance, in the UTD dataset, our model achieves comparable accuracy using approximately 60% of the data, with only a slight drop in performance relative to using 100% of the sequence. If this level of accuracy is deemed acceptable by radiologists, 60% of the data could serve as the stopping point. This decision would ultimately depend on the radiologists’ confidence in the model’s predictions and their willingness to adopt outputs based on fewer slices. The practical value of our approach lies in its robustness, as demonstrated on existing datasets. The model is able to maintain high performance even when fewer slices are used, making it a meaningful and flexible tool for real-world clinical workflows.

---

### Official Review · Reviewer_V1jg · 2024-11-03

**Soundness:** 3
**Presentation:** 3
**Contribution:** 2
**Rating:** 5
**Confidence:** 4

**Summary:**

The paper presents a method for sequential multiple instance learning in order to exploit the "sequential structure" of medical images. The proposed method uses a bidirectional transformer architecture to analyse these images, and outputs a sequence-aware uncertainty metric to indicate the certainty of the predictions. The proposed method is evaluated on one ultrasound and two CT datasets, and compared with three alternative baselines.

**Strengths:**

* Multiple instance learning is an interesting approach to the analysis of image sequence, and the proposal to do this with a sequence-aware method is a nice addition to the existing work.
* The uncertainty metric is a nice addition.
* The paper clearly describes the algorithm.

**Weaknesses:**

While the method might be interesting from a MIL perspective, I find it difficult to map this to a clinical use case. The assumptions behind the proposed model are not always clear, and the paper does not motivate how the sequential component of the method would work in practice.

1. The Introduction is fairly vague on what the "sequence" actually means: "each image study often contains multiple number of image instances". Does this mean a longitudinal study, or a single visit with multiple images? The context and practical application of the method remain unclear until the experiments.

2. The vague definition of a "sequence" makes it difficult to determine why the sequential structure would be better than a simple bag-of-instances approach. For the ultrasound: what does it mean for an instance to be earlier or later in the sequence? How were these scans obtained? Is there a given order in which these scans are usually made?

3. The arguments seem somewhat contradictory: on the one hand the paper claims that the sequence is important (we should use a sequence-aware method), but on the other hand the paper claims that we need a bidirectional method "to be robust to sequence reversals". For a CT scan, if it doesn't matter if we look from top-to-bottom or bottom-to-top, doesn't this suggest the sequence might be less important?

4. How does the subsequence approach work for CT scans? Isn't this very dependent on where the object of interest (i.e., the brain hemorrhage or pulmonary condition) appears in the scan? I don't really follow how the incremental prediction and uncertainty make sense here. If you only look at the subsequence without the object of interest, you'll find a completely different label.

5. The paper suggests that the method can be used to "terminate the imaging sequence" in order to "improve clinical efficiency and reduce radiation exposure". How would this work in practice? I might see how this could work for ultrasound (but this could be explained better: what does the "sequence" mean here? do the results depend on this order?), but for CT I find this a difficult claim. Usually, we make a single scan that collects all slices at the same time: you don't scan a few slices, analyse those, then "decide to stop or continue scanning".

Finally, the results are not compared with non-MIL methods for these datasets. Is the performance somewhat competitive? Do we really need a complex transformer for these tasks, or could we simply use a 3D convolutional network to classify these CT scans?

**Questions:**

Suggestions:

* Title/Text

  The title and parts of the paper refer to "clinical imaging". Isn't "medical imaging" a more commonly used term?

* Page #2 (Introduction):

  > is able to better capture difficult-toclassify, uncertain instances better

  Grammar: better ... better

* Page #3 (Sequential Multiple Instance Learning):

  > To further illustrate this concept, in Figure 2, we showcase the incremental prediction results

  I think this is supposed to refer to Figure 1? (See also the other references to Figure 2 in this section.)

* Page #4 (The BiSMIL Model and Training Process):

  > This is because scanning direction is usually a preference based on a particular clinician, and therefore we design our model to be robust to sequence reversals.

  What does this mean? Does this mean that the sequence of images might be presented in the reverse order, and that therefore the model should be direction invariant? If that is the case, can we trust the relative position of the scans, or is this also based on the clinician's preference?

* Page #6 (Dispersion and Sequence-Based Uncertainty):

  > Intuitively, if a sequence's incremental predictions quickly converge to 0 or 1, the model is more certain about that sequence. Conversely, if the predictions fluctuate significantly, the model is likely to be less certain about the predictions.

  I don't fully understand how this works with the bidirectional approach. Does this mean that if a sequence "converges" to 1, the same sequence in the reverse direction would converge to 0?

* Page #7 (Experiments):

  > For all of our experiments, we designated 70% of the data for training

  What does "the data" mean here? Samples? Scans? Images? Slices? Patients? I assume this is patients, but it would be good to make this more explicit.

  (If the data is not split on the patient level, this is something that would have to be fixed.)

* Page #7 (Experiments):
> UTD Classification Dataset

What is the nature of the sequence in these ultrasound scans? How are these scans generally made? What does it mean for an image to be early or late in the sequence?

* Page #7 (Experiments):

  > 50,862 brain CT slices

  What is the "sequence" for CT scans? Considering each "slice" as a separate step in the sequence would not make sense for me: the number of slices in a scan is determined before the scan is made, so the slice-by-slice MIL approach seems weird. I also find it difficult to see how this would work: would you order the slices from top to bottom, or would they appear in a random order in the sequence? If the order is sequential top-to-bottom, the ideal point to stop scanning would be the point where you first observe the hemorrhage, but how would that be done in practice?

* Page #8 (Final Prediction Accuracy):

  > which suggests that bidirectionality provides extra information that can improve the effectiveness of the model.

  In CT scans, this would mean that the slice direction is reversed. Does this cover a practical use case? Isn't the scanning direction usually given in the DICOM headers of the scan? (Or trivially determined from the image if this is not given?)

* Page #8 (Subsequence Prediction Accuracy):

  > To further understand the performance of our BiSMIL model, we compare the accuracy of the BiSMIL model against the three comparison models when only a subsequence of instances are revealed.

  How does this experiment work? Is this a random subset of the instances? How would this work in practice? (E.g., how do you make a random subset of CT slices?) Or is this a regular subsampling of the sequence, e.g., by taking every nth slice?

---

> ### Author Response · Authors · 2024-11-25
>
> We thank the reviewer comments and constructive criticisms and suggestions. Below, we address each point raised in the review.
>
> **1. Unclear Definition of “Sequence” and Its Practical Application**
>
> Thank you for pointing this out. In our paper, a “sequence” refers to the ordered set of images collected during a single visit or imaging session. For example, in CT scans, slices are typically acquired following a fixed anatomical direction (e.g., cranial-to-caudal). These anatomical orders inherently contain sequential information, which reflects spatial or contextual relationships crucial for diagnosis.
>
> Traditional MIL methods, which treat instances as unordered, are unable to capture this sequential information, limiting their ability to fully leverage the diagnostic potential of such imaging data. Our framework is designed to address this gap by explicitly incorporating the order of instances into the model.
>
>
> **2. Clarification on the Order in Medical Scans**
>
> We thank the reviewer for pointing this out. For ultrasound imaging, the scanning process typically follows a specific anatomical direction, such as the transverse plane or longitudinal plane. In this context, “earlier instances” refer to images captured earlier along the scanning trajectory, for example, starting from the left side in a transverse scan and moving toward the right.
>
> Most clinical scanning protocols adhere to such predefined directions, resulting in adjacent slices or frames that are more similar to each other, while changes observed in different regions along the sequence often provide critical diagnostic insights. The ordered structure thus captures valuable sequential information that not be considered in previous MIL frameworks.
>
> We will incorporate this explanation in the revised manuscript to clarify the role of order in medical imaging and its significance for diagnosis.
>
> **3. Sequence Importance vs. Bidirectionality**
>
> We thank the reviewer for raising this question, as it allows us to clarify our approach. The importance of sequence information lies in its ability to capture meaningful patterns, as described in 2. Clarification on the Order in Medical Scans: “Most clinical scanning protocols adhere to predefined directions, resulting in adjacent slices or frames that are more similar to each other, while changes observed in different regions along the sequence often provide critical diagnostic insights.”
>
> Bidirectionality addresses the need for robustness when interpreting the entire sequence as a whole. For example, whether the sequence is read from left to right or right to left (e.g., for CT scans), the diagnosis should remain consistent. This global direction reversal does not diminish the value of extracting sequential information from differences between adjacent slices, which remains critical for improving performance. This is why we adopt a bidirectional model architecture to balance robustness and the effective use of sequential information.
>
> **4. Subsequence and Incremental Training**
>
> In our paper, we emphasize the use of subsequence and incremental training to enable the model to learn the sequential information present among slices within a scan. During training, the model is always exposed to the full bag of data, as the subsequences from both directions complement each other. Furthermore, we ensure that the minimum subsequence percentage $\gamma \geq 0.5$ , which ensures that the model sees a substantial portion of the sequence during training.
>
> The incremental training approach allows the model to capture sequential patterns and make early predictions. For example, in the case of chest CT scans, this training strategy enables the model to achieve the same level of prediction accuracy using shorter subsequences during inference. This demonstrates the model’s ability to make efficient use of available information to generate reliable predictions.

---

> ### Author Response · Authors · 2024-11-25
>
> **5. Practical Use of Sub-sequence Prediction.**
>
> We appreciate the reviewer’s question and would like to elaborate further. As discussed in our response to Reviewer Azkw, the main advantage of achieving high accuracy with fewer instances is that doctors and radiologists can conclude scanning or imaging earlier if they are confident in the medical diagnosis. For example, assume that a doctor would consider the diagnosis reliable if the prediction accuracy exceeds 92%. As shown in Figure 4(a), the BiSMIL framework achieves this level of accuracy using approximately 60% of the sequence, compared to 90% required by the best competing method. This would allow radiologists to theoretically improve their workflow efficiency up to 1/60%=1.6x, and be able to scan more patients in a shorter period of time.
>
> While CT scans are typically acquired as a complete volume, the benefit of our framework lies in retrospective analysis, where fewer slices may need to be reviewed to reach a confident diagnosis. For ultrasound imaging, however, where images are acquired interactively, the sequential framework directly enables real-time termination of scanning when diagnostic confidence is achieved. This could significantly reduce imaging time and improve throughput.
>
> We hope our sequential framework can help alleviate the healthcare crisis by providing robust diagnostic evidence with fewer images, ultimately enhancing both clinical efficiency and patient outcomes.
>
> **6. Comparison with Non-MIL Methods**
>
> Thank you for this suggestion. We selected MIL methods as baselines because our task operates in a weakly supervised setting, where only bag-level labels are available. In contrast, non-MIL methods, such as 3D convolutional networks, generally require instance-level labels or fixed-size inputs, which are not only unsuitable for our problem setting but also more challenging to obtain in practice.
>
> **7. Clarifications on Specific Questions**
>
> • Title and Grammar (Page 2): We believe that “clinical imaging” better reflects the scope of issues discussed in this paper, and we will update the terminology consistently throughout the manuscript. Additionally, the grammar issue and the incorrect figure reference will be corrected in the final version.
>
> • Regarding the question on Page 4: We believe that this point has already been addressed in the explanations provided above.
>
> • Regarding the question on Page 6: When the incremental prediction of a sequence quickly converges to 0 or 1, it indicates that the sequence contains slices with highly decisive features for the label. For example, in the case of UTD, a specific scan might reveal a significantly enlarged urethral diameter, which is strongly indicative of the condition. Reversing the sequence does not change the model’s prediction, as explained earlier.
>
> • Regarding the question on Page 7: Yes, “patient” is the intended meaning here. We have also addressed the question about sequence order in our earlier responses.
>
> • Regarding the question on Page 8: Yes, we arrange the sequences based on the order of the DICOM series. For the Subsequence Prediction, we remove a certain proportion (e.g., 20%) of slices from the tail end of the ordered sequence to conduct the prediction.

---

> > ### Comment · Reviewer_V1jg · 2024-12-02
> >
> > I thank the authors for their replies to the reviewer comments. Based on their answers, I will maintain my previous rating.
> >
> > I appreciate the clarifications in the answers. However:
> > * I still have my doubts about the practical, clinical use case for the proposed method. I don't really see how this would be implemented in practice, and the paper doesn't explain this.
> > * I still would have appreciated a comparison with some state-of-the-art non-MIL baselines for these applications.
> >
> > > **5. Practical Use of Sub-sequence Prediction**
> > >
> > > the BiSMIL framework achieves this level of accuracy using approximately 60% of the sequence, compared to 90% required by the best competing method. This would allow radiologists to theoretically improve their workflow efficiency up to 1/60%=1.6x
> >
> > I would find this more convincing if the authors provided some justification. I think this estimate is very optimistic: even if you can reduce the number of images in the ultrasound sequence by 40%, this is unlikely to speed up the procedure with a similar amount (think set-up times, overhead etc.). Is there any way to estimate the real expected gains?
> >
> > > While CT scans are typically acquired as a complete volume, the benefit of our framework lies in retrospective analysis, where fewer slices may need to be reviewed to reach a confident diagnosis.
> >
> > I don't fully understand what this means. Who is doing the per-slice review? The automated classification system? Then it only saves some computation time. The radiologist? If so, how would this work in practice? They look at the first half of the scan and then stop?
> >
> > > **6. Comparison with Non-MIL Methods**
> > >
> > > We selected MIL methods as baselines because our task operates in a weakly supervised setting, where only bag-level labels are available. In contrast, non-MIL methods, such as 3D convolutional networks, generally require instance-level labels or fixed-size inputs, which are not only unsuitable for our problem setting but also more challenging to obtain in practice.
> >
> > Maybe I misunderstand the argument, but how is an "instance-level label" (I assume: one label for the whole CT scan?) different from a "bag-level label" (where the bag is also the whole scan?).
> >
> > I see why it is useful to compare with other MIL methods. However, I still think the paper would be more convincing if there was also a comparison with good non-MIL methods. If the argument is that the proposed method needs only 60% of the image data, then I would like to compare its performance with that of best alternative that uses the full image, not just the best alternative MIL method. (We don't know how good the MIL method is: it might be much worse than the state-of-the-art methods for these applications.)

---

> > > ### Author Response · Authors · 2024-12-03
> > >
> > > We thank the reviewer for their continued engagement and for raising important questions about the practical implementation and clinical utility of our method.
> > >
> > > Our approach is motivated by scenarios where reducing the number of slices required for accurate diagnosis can have significant benefits, including improving radiologists’ workflow efficiency and reducing patient exposure (in modalities involving radiation). For ultrasound, radiologists could terminate scanning earlier when the model predicts with high confidence, reducing unnecessary images. For CT scans, the method is intended for retrospective analysis, where a model can guide radiologists to focus only on diagnostically relevant slices, potentially reducing review time without requiring a complete manual assessment of all slices.
> > >
> > > In real-world implementation, the threshold for stopping or reducing the review workload would need to be determined in collaboration with radiologists, based on acceptable levels of accuracy and diagnostic confidence. We acknowledge that further validation and collaboration with clinical experts would be necessary to fully operationalize this framework in practice.
> > >
> > >
> > > >Maybe I misunderstand the argument, but how is an "instance-level label" (I assume: one label for the whole CT scan?) different from a "bag-level label" (where the bag is also the whole scan?).
> > > >
> > > An instance refers to each individual slice in a whole scan, while a bag refers to the entire scan composed of multiple slices. In some cases, it is possible to label each slice (instance) within a scan, but acquiring such instance-level labels is often prohibitively expensive and impractical for many clinical applications.
> > >
> > > In contrast, bag-level labels are more feasible and commonly available. These labels apply to the entire scan (bag), with the definition being based on the presence of positive instances within the bag. Specifically, in the Multiple Instance Learning (MIL) setting used in our work:
> > > - A scan (bag) is labeled as positive if it contains at least one positive slice (instance).
> > > - Conversely, it is labeled as negative if none of the slices (instances) are positive.
> > > We have described this definition in detail in Section 3.1 of the manuscript. This setup allows our model to leverage the weaker supervision of bag-level labels to learn patterns at the instance level, which is particularly useful in clinical settings where instance-level labeling is infeasible.

---

### Official Review · Reviewer_AZkv · 2024-11-06

**Soundness:** 3
**Presentation:** 3
**Contribution:** 2
**Rating:** 5
**Confidence:** 4

**Summary:**

This paper introduces the Sequential Multiple Instance Learning (SMIL) framework, which leverages the sequential nature of clinical imaging for improved early and final prediction accuracy. The authors propose a bidirectional Transformer model, BiSMIL, with a novel training procedure to utilize forward and reverse sequences effectively. Validated on multiple medical imaging datasets, BiSMIL outperforms existing methods in both final and early accuracy. Additionally, the SMILU uncertainty metric is introduced, offering sequence-aware uncertainty measures that better identify challenging cases. The work emphasizes the importance of sequence information in clinical imaging, with applications extending to other sequential data fields.

**Strengths:**

- **Interesting Problem:** The paper addresses a crucial issue in clinical imaging by introducing the SMIL framework, which effectively captures sequential information in diagnostic image analysis.

- **Clear Structure:** The paper is well-organized, with a logical flow that clearly explains the motivation, methodology, and results.

- **Comprehensive Experiments:** Detailed experiments and sequence length analyses demonstrate the model’s robustness and practical utility across multiple datasets.

**Weaknesses:**

- **Unclear Clinical Impact:** Although the authors suggest their approach could reduce patient radiation exposure, there is no comprehensive evaluation or discussion on this aspect. This lack of analysis raises questions about the practical clinical value and real-world applicability of the proposed method.

- **Limited Novelty in Methodology:** The proposed approach appears to mainly involve bidirectional input of subsequences with position encoding, which may not represent a substantial methodological innovation for addressing the ordered subsequence classification problem.

- **Lack of Visualization Analysis:** Given the use of a Transformer model, visualizing attention scores could provide a deeper understanding of the model’s internal mechanisms. This type of analysis would enhance interpretability and offer insights into how the model processes sequential information in clinical imaging data.

- **Overclaimed Interpretability:** The paper claims that the proposed method is interpretable; however, the uncertainty metric primarily supports quality control rather than true interpretability. The results demonstrate the metric's effectiveness in managing prediction quality, but further explanation is needed to clarify how and why the method enhances interpretability.

**Questions:**

- **Clarify Clinical Impact:** To strengthen the claim of reducing patient radiation exposure, the authors could provide a more detailed evaluation or discussion of this aspect. This might include an analysis of potential radiation savings in a clinical setting or simulations to illustrate practical benefits, thereby enhancing the method's real-world applicability.

- **Enhance Methodological Novelty:** To improve the perceived innovation, the authors could consider incorporating more advanced techniques tailored specifically for sequential classification challenges. Exploring unique modifications to the Transformer architecture, such as dynamic attention mechanisms adapted for clinical imaging sequences, may help distinguish their approach.

- **Add Visualization of Attention Scores:** Visualizing attention scores within the Transformer could provide valuable insights into the model's decision-making process and enhance interpretability. This would allow readers to see how the model weighs different parts of the sequence, helping to clarify the internal mechanics and providing a more comprehensive model analysis.

- **Clarify Interpretability Claims:** To support the interpretability claim, the authors could expand on why the uncertainty metric contributes to understanding the model's predictions. Providing examples or case studies where the metric aids in interpreting challenging cases would be helpful, or discussing how the metric reflects key decision points within the sequence, offering clinicians more insight into the model’s reasoning.

---

> ### Author Response · Authors · 2024-11-25
>
> We thank the reviewer for their detailed and constructive feedback. Below, we address the weaknesses and questions raised in the review.
>
> **1.Clarifying Clinical Impact**
>
> The main advantage of achieving high accuracy with fewer instances is that doctors and radiologists can conclude scanning or imaging earlier in the process if they are convinced about the medical diagnosis. For a simplified example, assume that the doctor would conclude the diagnosis is reliable if the accuracy hits above 92%. Then according to Figure 4(a),  the BiSMIL framework would only require the scanning of roughly 60% of the current sequence (compared to 90% for the best comparing method). This would allow radiologists to theoretically improve their workflow efficiency up to 1/60%=1.6x, and be able to scan more patients in a shorter period of time. This has significant real-world implications as currently there is a well-known shortage of radiologists across the globe, including in USA ([Radiology Workforce Shortage in the USA](https://www.acr.org/Practice-Management-Quality-Informatics/ACR-Bulletin/Articles/March-2024/How-Will-We-Solve-Our-Radiology-Workforce-Shortage#:~:text=There%20is%20a%20palpable%20shortage,on%20the%20ACR%20job%20board%20)), Canada ([Health Human Resource Crisis in Canada](https://www.newswire.ca/news-releases/addressing-the-health-human-resource-crisis-in-radiology-departments-across-canada-842727175.html)), and Norway ([Reporting Radiographers in Norway](https://www.sciencedirect.com/science/article/pii/S1078817423000329)). This has led to long wait times ([Wait Times for Imaging Services in Norway.](https://www.ncbi.nlm.nih.gov/pmc/articles/PMC10666297/#:~:text=Conventional%20X%2Dray%20(XR),7.9–11.4%20weeks%20respectively)), which in turn affects negatively patient outcomes ([Impact of Imaging Wait Times on Lung Cancer Prognosis](https://pubmed.ncbi.nlm.nih.gov/24931045/), [Reducing Lung Cancer Diagnosis Time](https://pubmed.ncbi.nlm.nih.gov/17975490/)).
>
> We hope that our sequential framework can help to combat this healthcare crisis by providing strong clinical evidence with fewer images, improving diagnostic efficiency.
>
> **2. Limited Methodological Novelty**
>
> While we acknowledge that position encoding and bidirectional processing are established techniques, the novelty of BiSMIL lies in its ability to systematically integrate sequential information into the MIL framework. To the best of our knowledge, no existing MIL methods effectively incorporate sequential information and can also provide early, accurate predictions. We believe that our work makes a significant and innovative contribution by offering an important insight: for many types of clinical imaging, utilizing the inherent sequential information is crucial for improved performance. Furthermore, we provide a comprehensive framework to address this challenge, including the development of a model architecture, a tailored training method, and an uncertainty metric to enhance interpretability and robustness.
>
> We appreciate the reviewer’s suggestions for further improvements, and we view them as promising directions for future exploration.
>
> **3. Visualization of Attention Scores**
>
> We thank the reviewer for pointing this out. We did not include specific visualizations of the attention scores within the Transformer because the model’s final predictions already involve computing attention scores over the features output by the Transformer and performing a weighted average. As shown in Figure 1, we illustrate an example where the attention scores align well with intuitive expectations—instances that contribute more significantly to determining the bag label are assigned higher attention scores.
>
> We believe that these attention scores already provide a clear understanding of the model’s prediction mechanism. However, we appreciate the suggestion and will consider including additional visualizations in the final version to further enhance interpretability.

---

> > ### Comment · Reviewer_AZkv · 2024-11-27
> >
> > I appreciate your further clarifications and efforts.
> >
> > However, for `3. Visualization of Attention Scores`: I am still confused as there are no specific explanations for the example in Figure 1 in both your response and your manuscript.
> >
> > For `1.Clarifying Clinical Impact`, I did not follow up with the authors' claim; previously, in their manuscript, they argued that reducing patient radiation exposure is good for patients' health. However, they are now arguing about the improved efficiency of the Radiologists. These two aspects are not very relevant from my perspective, which leads me to doubt the soundness of their motivation and clinical utility.
> >
> > I would be more convinced if further explanations were provided.

---

> > > ### Author Response · Authors · 2024-12-03
> > >
> > > >However, for `3. Visualization of Attention Scores`: I am still confused as there are no specific explanations for the example in Figure 1 in both your response and your manuscript.
> > > >
> > > We appreciate the reviewer’s request for further clarification regarding Figure 1. Referring to lines 141–146 in the manuscript, the scan sample in Figure 1 is taken from the UTD dataset, which consists of ultrasound images. A detailed description of the dataset can be found in Section 5.
> > >
> > > For UTD, radiologists typically determine the presence of the condition by identifying specific features within the urinary tract, such as dilation or abnormal structural characteristics. As shown in Figure 1, the three slices with higher attention values correspond to those that most prominently display features typical of UTD. This aligns with the radiologists' intuitive approach and their actual diagnostic workflow.
> > >
> > >
> > >
> > > >For `1.Clarifying Clinical Impact,` I did not follow up with the authors' claim; previously, in their manuscript, they argued that reducing patient radiation exposure is good for patients' health. However, they are now arguing about the improved efficiency of the Radiologists. These two aspects are not very relevant from my perspective, which leads me to doubt the soundness of their motivation and clinical utility.
> > > >
> > > We thank the reviewer for raising this point and would like to clarify that both reducing patient radiation exposure and improving the efficiency of radiologists are important benefits of our approach, stemming from its ability to make reliable decisions with fewer slices. These two aspects are interconnected but address different aspects of healthcare impact:
> > >
> > > - For patients: Reducing the number of slices required for diagnosis directly decreases radiation exposure in modalities like CT scans. Lower exposure is beneficial to patients' health, particularly for those requiring repeated scans, such as in follow-ups or chronic condition monitoring.
> > >
> > > - For radiologists and clinical workflow: Requiring fewer slices to make accurate predictions improves efficiency, allowing radiologists to spend less time on each scan. This increases the throughput of diagnostic workflows, enabling more patients to be scanned in the same amount of time. Given the global shortage of radiologists, this can significantly reduce patient wait times and improve healthcare delivery.

---

> ### Author Response · Authors · 2024-11-25
>
> **4. Overclaimed Interpretability**
>
> We thank you for the opportunity to clarify the concerns raised regarding the use of "interpretable" in our work. We recognize that interpretability typically implies that a model's decision-making process is transparent and logical from a human perspective. In our context, we employ the term to indicate that clinicians can discern whether the model's predictions align with their judgment and which instance is important for making the prediction. This definition of interpretability is akin to related literature like ([Early et al.,2022](https://arxiv.org/abs/2201.11701)) which defines interpretability for MIL. Figure 1 is designed to illustrate the model's incremental predictions in correlation with a clinician’s evaluation. For example, the first slice is identified as negative, whereas the second, fifth, and sixth slices exhibit positive characteristics, cumulatively leading to a positive final prediction. Thus, we use interpretability here as the ability of clinicians to understand how the model concludes its final prediction for the entire bag, thereby synchronizing their judgment with the model’s incremental predictions.
>
> Joseph Early, Christine Evers, and Sarvapali Ramchurn.Model agnostic interpretability for multiple instance learning.
> arXiv preprint arXiv:2201.11701, 2022.

---

### Meta-Review · Area_Chair_299b · 2024-12-21

**Metareview:**

The paper presents Sequential Multiple Instance Learning (SMIL), a framework leveraging sequential data in clinical imaging, utilizing a bidirectional Transformer (BiSMIL) for improved early and final diagnostic predictions, along with a sequence-aware uncertainty metric (SMILU). Strengths include its innovative application of MIL to sequential data, comprehensive experiments on medical datasets, and its effort to address efficiency and diagnostic accuracy challenges. However, reviewers noted significant weaknesses, particularly regarding the motivation for the sequential sequence of images; the claimed importance of sequence order is often not reflective of clinical reality, especially for modalities like CT and ultrasound where the order of images typically lacks diagnostic relevance. The proposed method's novelty was also questioned, as bidirectional architectures are well-established, and practical utility was unclear, with limited evidence of real-world applicability, such as how stopping imaging sequences in practice would be operationalized.

**Additional Comments On Reviewer Discussion:**

During the discussion, reviewers V1jg and Vuz2 raised concerns about the unclear motivation for the sequential sequence of images, particularly questioning its diagnostic relevance for modalities like CT and ultrasound, where order may not matter. EMs4 highlighted practical challenges in determining optimal stopping points for imaging sequences, while Vuz2 additionally questioned the absence of comparisons to non-MIL baselines like 3D CNNs. The authors provided clarifications, including examples of clinical scenarios leveraging sequence information and justifications for weakly supervised settings but struggled to convincingly address the relevance of sequential order and its practical implementation. Given the unresolved concerns about clinical applicability, methodological novelty, and the lack of strong empirical justification, the final decision of rejection was made given how most reviewers concerns still remained after the rebuttal.

---

### Decision · Program_Chairs · 2025-01-22

Reject